# Effects of High Intensity Plank Exercise on Physical Fitness and Immunocyte Function in a Middle-Aged Man: A Case Report

**DOI:** 10.3390/medicina57080845

**Published:** 2021-08-20

**Authors:** Sang-Kyun Park, Ki-Soo Lee, Seung-Jae Heo, Yong-Seok Jee

**Affiliations:** 1Department of Physical Education, Chungnam National University, Daehak-ro, Yuseong-gu, Daejeon 34134, Korea; ttl0033@gmail.com (S.-K.P.); lakasa2277@gmail.com (K.-S.L.); 2Department of Exercise Immunity for Cancers, Seoul Songdo Hospital, Dasan-ro, Jung-gu, Seoul 04597, Korea; 3Department of Leisure and Marine Sports, Hanseo University, Hanseo 1-Ro, Haemi-myeon, Seosan 31962, Korea

**Keywords:** plank exercise, physical fitness, muscle mass, CD56, cytotoxicity

## Abstract

*Background and Objectives*: Although the plank exercise is difficult to perform for untrained people, it does not require money, special equipment, or much space. However, it is not known how plank exercises affect physical fitness and immunocyte function. This study analyzed the changes in physical fitness and immune cells of a middle-aged man after performing 4 weeks of elbow plank exercise. *Materials and Methods:* Elbow plank exercise was performed for approximately 20 min (resting time, around 10 min) a day, 5 days a week for 4 weeks. The intensity was checked daily with ratings of perceived exertion (RPE). When the participant reached an intensity of RPE 15, RPE 16, and RPE 17 of the RPE 20 scale, 1 min of rest was given before repeating the process. *Results:* Compared with the pre-values, (1) all the physical fitness factors (muscle strength, muscle endurance, flexibility, and cardiopulmonary fitness) were increased after 4 weeks; (2) body weight, skeletal muscle mass, and the basal metabolic rate were increased, whereas body fat mass and fat percentage were decreased; and (3) although the variables of complete blood count were changed positively, some were not. Specifically, CD3, CD8, and CD56 were increased, whereas CD4, CD4/CD8, and cytotoxicity were decreased. These results show that elbow plank exercise can improve all factors of physical fitness and improve some of the immunocyte functions of a middle-aged man. *Conclusions:* This study confirmed that, although the elbow plank exercise of vigorous intensity for 4 weeks improved physical fitness, it was not effective in improving some immunocyte functions. Therefore, the exercise intensity of plank exercises for improving immunocyte functions should be reconsidered.

## 1. Introduction

The Korea Disease Control and Prevention Agency reported that 35.4% of the nation’s population lacked physical activity (PA) [1]. This lack of PA is of increasing interest because it also affects the high mortality rate [2]. Recently, social distancing has become mandatory due to the COVID-19 pandemic, so outdoor activities have decreased, and sedentary lifestyles have increased, resulting in a relative decrease in the amount of PA. In most cases, but not in all cases, lack of PA leads to obesity and impairs the function of immune cells to defend against foreign cells, such as the coronavirus [3].

Exercise is a helpful way to reduce body weight and can prevent obesity by maintaining and/or increasing the muscle mass and basal metabolic rate [4]. Moreover, it can improve immune cell function [5,6,7,8,9,10,11]. In other words, during the present COVID-19 pandemic, exercise is a necessity, not an option [12]. Considering the recent mandatory social distancing, it is important to find and perform exercises that can be done alone at home [13]. Recently in Korea, home training is trending since it allows individuals to exercise by themselves in the comfort of their own homes. Since it requires little time, space, and cost, many studies are being conducted on its efficacy [14].

Plank exercise is a movement derived from Pilates, yoga, and stretching, and can be performed by anyone with minimal spatial requirements. Since plank exercise can be used as a whole-body exercise, it is possible to expend a large amount of calories and develop muscles in a short period of time [14]. Some studies have shown that plank-type exercises are effective in developing strength and endurance [15,16], in reducing low back pain [17], and in preventing falls [18]. Though past studies have shown plank exercise to be effective in engaging the core muscles of the human body, it is not known what kind of change it provides to the immune cells of the human body. Therefore, this study was conducted to observe changes in physical fitness and immunocyte function in a male subject after performing plank exercises at home.

## 2. Materials and Methods

### 2.1. Participant

A 43 year-old male, who had no experience performing plank exercises, voluntarily participated in the study. Before the start of the experiment, his height and body weight were measured at 168.6 cm and 80.4 kg, respectively. He played basketball on a regular basis 10 years ago, but now his main physical activity is only walking for 30 min twice a day. He had no cardiac or metabolic disease, but his father had hypertension. He has been smoking a pack of cigarettes a day and drank moderately about three days a week. He had not been taking any medication. Of course, we recommended abstaining from exercise, smoking, and alcohol consumption for one month during the experiment.

### 2.2. Experimental Design

This prospective case study compared pre-values with post-values and was conducted at Seoul Songdo Hospital from May 28 to 26 June 2021. It followed the principles of the Declaration of Helsinki and received approval from the institutional ethics committee (Sahmyook Univ. 2-1040781-A-N-012020085HR). Prior to the study, the principal investigator explained all the procedures to the participant before he read and signed an informed consent form. He completed a self-report questionnaire about his health status and learned how to record the ratings of perceived exertion (RPE) in a diary. The assessments were performed at Week 0 (baseline) and at Week 4. Plank exercises were used as the intervention program, which were conducted for 4 weeks, 5 days a week, for 20 min (+ a rest time of 10 min) a day. The physical fitness variables measured were body composition, muscle strength, muscle endurance, flexibility, and cardiopulmonary fitness. Other dependent variables included complete blood count (CBC), lymphocytes, and granulocytes.

### 2.3. Measurement Methods

#### 2.3.1. Blood Sampling and Immunocyte Measures

This study investigated the changes of CBC, lymphocyte, and granulocytes subsets. The percentage (%) and absolute cell counts of peripheral blood cell subsets were analyzed as described below: 50 μL of blood were stained with anti-human antibodies against anti-CD3, anti-CD4, anti-CD8, and anti-CD56 from BD Biosciences (Franklin Lakes, NJ, USA). After incubation for 15 min at room temperature (RT) in the dark, the red blood cells (RBC) were lysed by adding 450 μL of FACS lysing solution to each test tube for 15 min at RT in the dark. Another intracellular staining step was required for analysis of regulatory T cells, cytotoxic T cells, and immune checkpoint molecules. After the lysis of RBC, the remaining cells were washed in 2 mL of permeabilization buffer. After staining was completed, cells were analyzed using FACS Canto II (BD Bioscience) and Flowjo software (Treestar, Ashland, OR, USA) and are presented as percentages. Absolute cell counts of the lymphocyte subsets were obtained using an automatic hematology analyzer (Sysmex Corp., Kobe, Japan). The analyzed CBC subsets consisted of white blood cells (WBC), RBC, hemoglobin, hematocrit, platelets, mean corpuscular volume (MCV), mean corpuscular hemoglobin (MCH), mean corpuscular hemoglobin concentration (MCHC), erythrocyte sedimentation rate (ESR), red cell distribution width (RDW), and platelet distribution width (PDW). Percentage analysis for granulocytes were composed of neutrophils, lymphocytes, monocytes, eosinophils, and basophils. The analyzed immunocytes were a lymphocyte subset of immunophenotypes, which were CD3, CD4, CD8, CD56, and CD4/CD8. The cytotoxicity refers to the volume of natural killer (NK) cells against viral, bacterial, or cancerous cells.

#### 2.3.2. Physical Fitness Measures

In this study, the health-related physical fitness components included body composition, muscle strength, muscular endurance, flexibility, and cardiopulmonary endurance, which was measured using a graded exercise test. Firstly, body composition was measured using a bioelectrical impedance analysis method with a body composition analyzer (Inbody 770, Biospace, Seoul, Korea). The variables of body composition in this study were height, weight, skeletal muscle mass, fat mass, body mass index (BMI), fat percentage, and basal metabolic rate (BMR) [19]. Secondly, muscle strength was measured using a grip strength test with a Smedley dynamometer (TKK-5401, Takei Inc., Tokyo, Japan). The subject held the dynamometer while his arms did not contact the body. After both hands were alternately measured twice, the maximum value was recorded, and the mean value from both hands was used [20]. Thirdly, muscle endurance was measured using a sit-up test for 60 s [21]. The subject lay down with his back on the floor, bended his knees at right angles, fixed his feet on the sit-up board, and placed his hands behind his head with his fingers crossed. The total number of completed sit-ups was recorded. Fourthly, flexibility was measured using a sit-and-reach test that measured the degree to which the upper body bends forward in a sitting position with both legs fully outstretched. The subject took off his shoes and sat with his knees straightened before bending his upper body forward and extending his head toward the scale above a flexibility meter (TKK1859, Takei Inc., Tokyo, Japan). The maximum value of two measurements was recorded [22]. Lastly, this study assessed the maximal oxygen uptake (VO_2_max) for cardiopulmonary fitness using a graded exercise test. The devices used included an electrocardiogram (Q-4500, SunTech Medical, Inc., Morrisville, NC, USA), automatic sphygmomanometer (M-412), gas tester (QMC4200), and treadmill (Q65-90, Quinton, New Kent County, VA, USA). The Bruce protocol was used, which consisted of Stage 1 (5 metabolic equivalents; METs), Stage 2 (7 METs), Stage 3 (9 METs), Stage 4 (11 METs), and subsequent stages that followed similar increases. The subject continued to walk or run until reaching an all-out level, which is their maximal RPE. The VO_2_max was calculated using body weight at a peak test stage [23].

### 2.4. Exercise Program

Elbow plank exercise was performed for the study. Lying stretches were performed for 5 min before and after the plank exercise. In the work-out phase, the participant maintained a straight, strong line from head to toes with no lowering of the hips with the shoulders and elbows flexed at 90°, as shown in Figure 1.

In this study, Borg’s 20-scale RPE chart for deciding his plank intensity was copied to the size of the palm of the hand and provided to the participant. By referring to various studies in the literature [14,24,25], the plank exercise was conducted with the goal of reaching an extremely high intensity. The exercise intensities for elbow plank ranged from RPE 15 (feeling hard) and RPE 17 (feeling extremely hard). Each 30-min session was divided into three 10-min stages, each with increasing intensity. In other words, elbow plank exercises were performed until reaching RPE 15 (hard) in the first 10 min, RPE 16 (hard to very hard) in the second 10 min, and RPE 17 (very hard) in the third 10 min. The same posture was maintained until reaching the target RPE for each stage. When the target RPE was reached, the subject took rested for 1 min and then restarted throughout the approximate 30-min duration. In other words, if each intensity was exceeded and the correct plank posture was not achieved, 1 min was allowed to rest, to then re-execute.

As shown in Table 1, the average plank exercise time performed by the subject in the first week was 17.22 min, and the rest time was 14 min, making the total program 31.22 min long. On the other hand, the average plank exercise time of Week 2 was 16.81 min, and the rest time was 13.4 min, giving a total of 30.21 min. The average plank exercise time of the 3rd week was 17.68 min, and the rest time was 13 min, for a total of 30.68 min. Finally, the average plank exercise time of Week 4 was 18.07 min, and the rest time was 12.6 min, which gave a total of 30.67 min. That is, in this study, the set time at RPE 15 and RPE 16 was adjusted to match the total program time to around 30 min, and the time at RPE 17 was not adjusted for high-intensity plank exercise.

### 2.5. Data Analyses

Microsoft Excel (Microsoft, Redmond, WA, USA) was used to organize the data. In order to observe changes before and after the plank exercises, the delta percentage (Δ%) was calculated using the formula of ‘{(post data − pre data)/pre data} × 100’ for all data.

## 3. Results

### 3.1. Effect of Plank Exercise on Complete Blood Count

As shown in Table 2, the WBC, RBC, hemoglobin, hematocrit, and platelets increased, whereas MCV, MCH, and MCHC decreased after 4 weeks. Meanwhile, the ESR showed no changes after 4 weeks of plank exercise. In addition, although the RDW decreased, the PDW increased. These results indicate that elbow plank exercise may change the blood components in middle-aged men after 4 weeks.

### 3.2. Effect of Plank Exercise on Lymphocytes and Granulocytes

As shown in Table 3, although the neutrophil and basophil increased, the lymphocytes, monocytes, and eosinophils decreased after 4 weeks. Meanwhile, although the CD3, CD8, and CD56 increased, the CD4, CD4/CD8, and cytotoxicity decreased. These results indicate that elbow plank exercise may affect the immunocyte function in middle-aged men. In particular, when looking at the changes in NK cell-related functions, it appears that the plank exercises led to notable improvements after just 4 weeks.

### 3.3. Effect of Plank Exercise on Body Composition

As shown in Table 4, body weight, skeletal muscle mass, BMI, and BMR increased, whereas body fat mass and body fat percentage decreased after 4 weeks. These results show that elbow plank exercise can change the body composition of middle-aged men. In particular, performing plank exercises for 4 weeks showed an increase in total body weight due to an increase in skeletal muscle mass (3.10%) and a change in body fat mass (−2.98%).

### 3.4. Effect of Plank Exercise on Physical Fitness Levels

As shown in Table 5, muscle strength, muscle endurance, flexibility, and maximum oxygen uptake increased after 4 weeks of plank exercise. These results indicate that 4 weeks of elbow plank exercise can improve the health physical fitness of middle-aged men.

## 4. Discussion

This study found that elbow plank exercise improved the body composition and increased the health-related physical fitness in a middle-aged man, leading to desirable changes in immunocyte function. In the health-related fitness results of this study, plank exercise was associated with a decreasing tendency in body weight, including fat levels, and maintaining or enhancing skeletal muscle mass and the basal metabolism rate. Furthermore, this study found that the body weight, skeletal muscle mass, BMI, and BMR increased, whereas body fat mass and fat percentage decreased after 4 weeks. These results show that plank exercise can improve the body composition of middle-aged men. In particular, 4 weeks of elbow plank exercises led to an increase in total body weight due to an increase in skeletal muscle mass (+3.10%) and a change in body fat mass (−2.98%).

Innate immunity involves macrophages, neutrophils, dendritic cells, and NK cells. The acquired immune response involves B cells and T cells (CD4+ and CD8+) [26]. In the aspect of immunocyte functions in this study, although the neutrophils, basophils, and NK cells (CD56) increased, the lymphocytes, monocytes, and eosinophils decreased after 4 weeks. Meanwhile, the total T cell (CD3) and cytotoxic T cell (CD8) increased, while the helper T cell (CD4), CD4/CD8, and cytotoxicity decreased. These results show that ‘hard’ to ‘very hard’ plank exercise changed the immunocyte functions in a middle-aged man. In particular, when looking at the reduced cytotoxicity related to NK cell functions, it appears that 4 weeks of plank exercises were effective.

It has been reported that exercise should be a part of a treatment program for treating chronic disease. Recently, the coronavirus has disrupted the immunocytes of many people as a result from mid- to long-term isolation [12]. In this respect, the results of this study are considered to be meaningful for the current global situation. The purpose of this study was to understand how the plank exercise, an exercise that can be performed alone at home, affects the immunocyte function and physical strength of a single subject. For immunocyte function, Pedersen [27] reported that exercise influences both innate and acquired immunity. Inkabi et al. [28] reported that different immunocyte types are affected differently by physical exercise. In this study, the lymphocyte percentage, which is related to innate immunity, somewhat decreased from the baseline (34.8%) to Week 4 (34.5%). Moreover, similar to lymphocyte percentage, this study observed a reduction in NK cell percentage and cytotoxicity percentage. This result indicates that although regular exercise leads to positive changes in innate immune function, vigorous exercise may cause negative changes. Previously, Gleeson [29] and Pedersen and Hoffman-Goetz [30] reported that NK cells increase in abundance during exercise, while the NK cell count drops to below half of the normal level after exercise. Del Giacco et al. [26] also indicated that an increase in NK cells, as measured as a percentage of lymphocytes in peripheral blood, enhances cytolytic capacity. A decrease in the level of NK cells results in suppression of cytolytic activity, which may indicate an enhanced period of susceptibility to infections [31]. NK cells are remarkably sensitive to the stress induced by physical exercise, which promotes their redistribution from the peripheral blood to other tissues after physical exercise [26]. There is an increase in the number of NK cells in the peripheral blood that are transported to other tissues during physical exercise due to induced stress signals; the target tissues should be reached before the cessation of physical exercise, with the blood serving as a highway to traffic NK cells to the sites of stress signaling [32]. Similarly, the results of this study showed that the NK cell levels, which are related to innate immunity, tended to decrease with exercise. In other words, the hard to intense plank exercise in this study did not change the levels of NK cells, whereas the levels of cytotoxicity in the peripheral blood decreased due to overstressed signals. Similar to the results of this study, Pedersen [27] reported that a moderate amount of exercise provides an overall “boost” to the immune system, but strenuous exercise results in dampening of the immune system.

Previous studies suggest that physical exercise is important for innate immune function, but more important for acquired immune function. Taking the above studies into account, Fabbri et al. [33] showed that a crucial consequence of defective T cell function is an increased incidence of viral infections. B cells produce antibodies, which are released to destroy invading viruses and bacteria [34]. CD4+T cells function to activate cells of the innate immune system, such as B lymphocytes and cytotoxic T cells. CD4+T cells are also involved in the suppression of immune reactions [35]. Similarly, the results of this study showed that the helper T cell count (CD4), a marker of adaptive immunity, decreased in the participant, whereas the CD4/CD8 ratio decreased −25.76% at Week 4 compared to the baseline. In other words, regular plank exercise produced a decrease in the helper T cell population while suppressing cytotoxic immune cells. In addition, this study showed that the CD8 T cell percentage increased. These changes in adaptive immune cells significantly increased cytotoxicity, revealing that the apoptotic activity of NK cells and WBC can actively contribute to the killing of bad cells. During physical exercise, CD4+ and CD8+T cells, also known as helper and cytotoxic T cells, respectively, were recruited to the peripheral blood, resulting in increased concentrations of lymphocytes. In general, it is known that the functions of B cells are suppressed after intense, long-duration exercise, while lymphocyte concentrations have been shown to increase during acute exercise and fall below the pre-workout values after long-term endurance exercise [26,36]. However, Pedersen and Nieman [37] reported that the total lymphocyte concentration declines after acute exercise and the proliferation response is unchanged compared to the response before exercise. Physical exercise induces a greater early increase in catecholamines that affects different types of lymphocytes, resulting in their mobilization in the blood [38]. In other words, this study implies that changes in immunocyte function occur simultaneously with changes in fitness-related variables.

Strenuous exercise, but not moderate exercise, is followed by a decreased concentration of lymphocytes in the bloodstream, which results in low lymphocyte levels in tissues [39]. In light of these results, we can infer that the plank exercise performed in this study was maintained at a high intensity by the participant. In the aspect of physical fitness, the strength, muscle endurance, flexibility, and VO_2_max were increased after 4 weeks of plank exercise. These results indicate that plank exercise can improve health-related physical fitness in middle-aged men, though it seems that this plank exercise made negative changes in NK cell-related functions due to the high intensity. In other words, the results of this study showed that high-intensity plank exercise can increase the number of NK cells but reduce its cytotoxicity. Normally, when the number of NK cells increases, cytotoxicity also increases, but cytotoxicity decreases when there is a specific cause, such as having cancer cells, when exposed to excessive stress, or when exercising excessively [40,41]. In other words, it can be concluded that the subject who completed 4 weeks of plank exercises had increased NK cells, but showed no changes in its function due to the negative results of high-intensity exercise. Meanwhile, immune cells, as well as physical fitness, were measured one day before the start of the plank exercises and on the day after the four-week period. As a result, a clear change could be observed in the physical fitness variables, but a constant change pattern could not be observed in the variables that had sensitive responses, such as immune cells. Therefore, based on the data derived from the current case study, we suggest that future studies should not only look at the changes before and after the experiment to observe the changes in immune cells but rather in the middle of the course of the experiment, 7 days after the experiment, or one month after the experiment.

## 5. Conclusions

This study confirmed that high- to very-high-intensity plank exercise can improve immunocyte function and physical fitness in a healthy man. However, our study has some limitations. First, the participant consisted entirely of only one middle-aged man, which is a small sample size. Second, although there are hundreds of types of immune cells, this study only observed a few specific immunocytes. Third, a careful approach is required because plank exercises performed at less than moderate intensity or high intensity can cause shoulder joint or back pain. Considering these limitations, further studies that investigate the effectiveness of plank exercise on a greater number of participants with diverse demographic backgrounds, and on multiple immune cell tests, are encouraged.

## Figures and Tables

**Figure 1 medicina-57-00845-f001:**
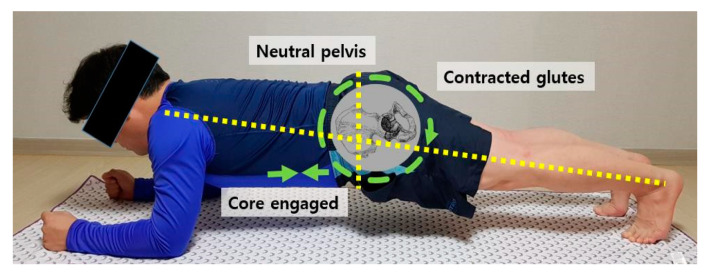
Elbow plank exercise at home.

**Table 1 medicina-57-00845-t001:** Plank exercise time and resting time of the participant.

	Days	Plank Exercise		Resting Time		Total (Minutes)
		Seconds	Minutes	Seconds	Minutes	
Week 1	Day 1	1079.00	17.98	840.00	14	31.98
	Day 2	1049.00	17.48	840.00	14	31.48
	Day 3	1026.00	17.10	840.00	14	31.10
	Day 4	1028.00	17.13	840.00	14	31.13
	Day 5	985.00	16.42	840.00	14	30.42
	Mean		17.22		14.00	31.22
Week 2	Day 1	1052.00	17.53	840.00	14	31.53
	Day 2	1034.00	17.23	780.00	13	30.23
	Day 3	1023.00	17.05	780.00	13	30.05
	Day 4	983.00	16.38	780.00	13	29.38
	Day 5	951.00	15.85	840.00	14	29.85
	Mean		16.81		13.40	30.21
Week 3	Day 1	1108.00	18.47	780.00	13	31.47
	Day 2	1051.00	17.52	780.00	13	30.52
	Day 3	1066.00	17.77	780.00	13	30.77
	Day 4	1079.00	17.98	780.00	13	30.98
	Day 5	999.00	16.65	780.00	13	29.65
	Mean		17.68		13.00	30.68
Week 4	Day 1	1071.00	17.85	720.00	12	29.85
	Day 2	1085.00	18.08	780.00	13	31.08
	Day 3	1029.00	17.15	720.00	12	29.15
	Day 4	1106.00	18.43	780.00	13	31.43
	Day 5	1130.00	18.83	780.00	13	31.83
	Mean		18.07		12.6	30.67

**Table 2 medicina-57-00845-t002:** Changes in complete blood counts.

	Baseline	Week 4	Δ%
White blood cell (×10^3^/μL)	6.1	6.7	9.84
Red blood cell (×10^6^/μL)	4.8	5.1	6.25
Hemoglobin (g/dL)	15.4	16.2	5.19
Hematocrit (%)	43.4	45.9	5.76
Platelets (×10^3^/μL)	266	274	3.01
Mean corpuscular volume (fL)	89.9	89.5	−0.44
Mean corpuscular hemoglobin (pg)	31.9	31.6	−0.94
Mean corpuscular hemoglobin concentration (g/dL)	35.5	35.3	−0.56
Erythrocyte sedimentation rate (mm/hr)	4	4	0
Red cell distribution width (%)	11.6	11.4	−1.72
Platelet distribution width (%)	13.4	16.6	23.88

All values are expressed as original data. Δ% means changed ratio, which get from {(post data − pre data)/pre data} × 100.

**Table 3 medicina-57-00845-t003:** Changes in granulocytes and immunocytes.

	Baseline	Week 4	Δ%
Neutrophil (%)	53.9	55.7	3.34
Lymphocyte (%)	34.8	34.5	−0.86
Monocyte (%)	7.7	6.4	−16.88
Eosinophil (%)	3.3	3	−9.09
Basophil (%)	0.3	0.4	33.33
CD3 (%)	67.5	68	0.74
CD4 (%)	38.4	33.7	−12.24
CD8 (%)	29.1	34.3	17.87
CD56 (%)	7.26	7.33	0.96
CD4/CD8 (%)	1.32	0.98	−25.76
Cytotoxicity (%)	25.54	7.71	−69.81

All values are expressed as original data.

**Table 4 medicina-57-00845-t004:** Changes in body composition.

	Baseline	Week 4	Δ%
Weight (kg)	80.4	81.5	1.37
Skeletal muscle mass (kg)	32.3	33.3	3.1
Body fat mass (kg)	23.5	22.8	−2.98
Body mass index (kg/m^2^)	28.6	28.7	0.35
Body fat percentage (%)	29.2	28.6	−2.05
Basal metabolism rate (kcal)	1599	1638	2.44

All values are expressed as original data.

**Table 5 medicina-57-00845-t005:** Changes in health physical fitness components.

	Baseline	Week 4	Δ%
Muscle strength (kg)	41.75	42.65	2.16
Muscle endurance (reps)	28	35	25
Flexibility (cm)	−6	3	150
VO_2_max (ml/kg/min)	35.6	36.3	9.84

All values are expressed as original data.

## Data Availability

Data and material are available on reasonable request.

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
