# Peer review of "Effects of High Intensity Plank Exercise on Physical Fitness and Immunocyte Function in a Middle-Aged Man: A Case Report"

_medicina, 2021, doi:10.3390/medicina57080845_

Round 1
Reviewer 1 Report
The manuscript is of interest, but shows serious flows that need to be addressed. there are a large number of unsubstantiated statements that need to be supported by references.
Title: this is a case report, rather than a case study. Please change.
Abstract: Plank exercise is anything but easy. It is indeed quite difficult and challanging to perform it correctly for untrained people.
Introduction:
l. 32 what it meant by "lack of PA"? Please be precise.
l. 36 provide reference
l. 36. obesity is a multifactorial disease, it is an unacceptable oversiplification to say "lack of physical activity causes obesity". Please correct.
l 38-39 "Exercise is an effective way to reduce body weight and can effectively prevent obe-38 sity by increasing muscle mass and basal metabolic rate." this again is a large oversimplification and is not supported by any referece. Correct and provide reference.
l. 39-40 "it can improve immune cell function" please provide reference
l. 40-41 "during the present COVID-19 pandemic, exercise is a 40 necessity, not an option" - this is true also for non-Covid-19 times
l. 44 provide reference
l. 44 Plank exercise is far from being easy
l. 45 provide references
l. 49 provide reference
l 49-54 make sure all references relate explicitely to plan exercise
l. 55-56 "there are many people who 55 do not meet the vaccine investigation for coronavirus" - what is this supposed to mean?
Methods
l. 79-80 explain why these outcomes were selected
l. 114 add manufacturer and data on validity and test-retesr
l. 117 sit-up test for 60 sec - add reference and data on validity and test-retest
l. 145-146 By referring to various 145 literatures - add reference specific to plank exercise
l. 152-154 it is absolutely necessary to provide more details on the progression of the training. Plank exercise is indeed very taxing, and it is of interest how the participant was able to perform at the beginning and how he progressed. Please give details in minutes, not in seconds
It would have been of much interest to have more than two data assessment time points.
Results
Table 4.
the change in flexibility does not seem realistic.
Discussion
l. 202 quality of life is not the subject matter of this report, and the statement as such is not true. Please delete or rewrite in a manner that is it true, provide reference.
l. 220 "Previous studies have found both positive and negative effects of regular exercise in patients with various types of diseases". - this sentence is empty, and unscientific. Please be precise and use appropriate references
l 223 add reference
l. 289 add reference
Whole of the discussion: please make sure to differentiate between endurance vs. strength training in terms of immune response.
Limitations
A limitation of the study is the questionable generalisability of this protocol. Plank exercise is indeed very challenging, and if performed incorrectly can induce low back pain or shoulder pain. This should be also discussed.
Author Response
Answers to 1st reviewer’s comments
Thank you for your kind advice and comments for publication in Medicina. We revised our manuscript as per your comments. We represented the specific modifications in response to the comments by blue-letters in our manuscript. We sincerely appreciate your comments because your comments make our manuscript better. Details of responses about reviewer’s comments are as follows:
#1. Comments or Suggestions: The manuscript is of interest, but shows serious flows that need to be addressed. there are a large number of unsubstantiated statements that need to be supported by references.
Response: First of all, thank you for your comments. Based on your comments and comments, I've attached references to unsubstantiated statements. Thank you for taking a look at the following.
#2. Comments or Suggestions: Title: this is a case report, rather than a case study. Please change.
Response: Based on your suggestion, 'case study' has been changed to 'case report'.
Abstract:
#3. Comments or Suggestions: Plank exercise is anything but easy. It is indeed quite difficult and challanging to perform it correctly for untrained people.
Response: We changed the above sentence to 'Although the plank exercise is difficult to perform for untrained people, it does not require money, special equipment, or much space.' according to your comment.
Introduction:
#4. Comments or Suggestions: l. 32 what it meant by "lack of PA"? Please be precise.
Response: According to your comment, we changed physical activity to ‘PA’.
#5. Comments or Suggestions: l. 36 provide reference
Response: According to your comment, we attached below reference in the end of sentence as a number.
[3] Nieman, D.C.; Wentz, L.M. The compelling link between physical activity and the body's defense system. Journal of Sport and Health Science, 2019, 8, 201-217.
#6. Comments or Suggestions: l. 36. obesity is a multifactorial disease, it is an unacceptable oversiplification to say "lack of physical activity causes obesity". Please correct.
Response: After researching several documents and thinking about it, it seems that your opinion is correct. So, based on your comments, we have revised that sentence to the following sentence: “In most cases, but not in all cases, lack of PA leads to obesity… ‘
#7. Comments or Suggestions: l 38-39 "Exercise is an effective way to reduce body weight and can effectively prevent obesity by increasing muscle mass and basal metabolic rate." this again is a large oversimplification and is not supported by any referece. Correct and provide reference.
Response: According to your comment, we corrected the sentence and attached the reference in the end of sentence as a number. "Exercise is a helpful way to reduce body weight and can prevent obesity by maintaining and/or increasing muscle mass and basal metabolic rate."
[4] Stiegler, P.; Cunliffe, A. The Role of Diet and Exercise for the Maintenance of Fat-Free Mass and Resting Metabolic Rate During Weight Loss. Sports Medicine, 2006, 36, 239–262.
#8. Comments or Suggestions: l. 39-40 "it can improve immune cell function" please provide reference
Response: According to your comment, we attached the reference in the end of sentence as a number.
[5] Jee, Y. S. Exercise is an antigen for vaccination: first series of scientific evidence. J Exerc Rehabil. 2019, 15(3), 339-340.[6] Jee, Y. S. How much exercise do we need to improve our immune system?: second series of scientific evidence. J Exerc Rehabil. 2020, 16(2), 113-114.[7] Jee, Y. S. Influences of acute and/or chronic exercise on human immunity: third series of scientific evidence. J Exerc Rehabil. 2020, 16(3), 205-206.[8] Jee, Y. S. Physical exercise for strengthening innate immunity during COVID-19 pandemic: 4th series of scientific evidence. J Exerc Rehabil. 2020;16(5):383-384.[9] Jee, Y. S. Acquired immunity and moderate physical exercise: 5th series of scientific evidence. J Exerc Rehabil. 2020, 16(5), 383-384.[10] Jee, Y. S. Cancer and exercise immunity: 6th series of scientific evidence. J Exerc Rehabil. 2021, 17(3), 151-152.[11] Peake, J.; Nosaka, K.; Suzuki, K. Characterization of inflammatory responses to eccentric exercise in humans. Exerc Immunol Rev 2005, 11, 64-85.
#9. Comments or Suggestions: l. 40-41 "during the present COVID-19 pandemic, exercise is a necessity, not an option" - this is true also for non-Covid-19 times
Response: According to your comment, we attached the reference in the end of sentence as a number.
[12] Nyenhuis, S.M.; Greiwe, J.; Zeiger, J.S.; Nanda, A.; Cooke, A. Exercise and Fitness in the age of social distancing during the COVID-19 Pandemic. J Allergy Clin Immunol Pract, 2020, 8, 2152-2155.
#10. Comments or Suggestions: l. 44 provide reference
Response: According to your comment, we suggested the reference as below and attached the reference number in the end of sentence.
[13] Schultz, D.; Jones, S.S.; Pinder, W.M.; Wiprovnick, A.E.; Groth, E.C.; Shanty, L.M.; Duggan, A. Effective Home Visiting Training: Key Principles and Findings to Guide Training Developers and Evaluators. Matern Child Health J. 2018, 22, 1563-1567.
#11. Comments or Suggestions: l. 44 Plank exercise is far from being easy
Response: According to your comment, we changed the sentence as follows. “Since it requires little time, space, and cost, many studies are being conducted on its efficacy.”
#12. Comments or Suggestions: l. 45 provide references
Response: According to your comment, we suggested the reference as below and attached the reference number in the end of sentence.
[14] Park, D.J.; Park, S.Y. Which trunk exercise most effectively activates abdominal muscles? A comparative study of plank and isometric bilateral leg raise exercises. J Back Musculoskelet Rehabil. 2019, 32, 797-802.
#13. Comments or Suggestions: l. 49 provide reference
Response: According to your comment, we suggested the reference as below and attached the reference number in the end of sentence. Changed sentence is follows: ……Since plank exercise can be used as a whole-body exercise, it is possible to expend a large amount of calories and develop muscles in a short period of time [14].
[14] Park, D.J.; Park, S.Y. Which trunk exercise most effectively activates abdominal muscles? A comparative study of plank and isometric bilateral leg raise exercises. J Back Musculoskelet Rehabil. 2019, 32, 797-802.
#14. Comments or Suggestions: l 49-54 make sure all references relate explicitely to plan exercise
Response: Based on your opinion, we have investigated all of them (below 4 references) and found references that are relevant to our intended meaning. Also, for clearer expression, the sentence has been modified as follows. … the plank type’s exercises are effective in developing strength and endurance…
- Akuthota, V.; Ferreiro, A.; Moore, T.; Fredericson, M. Core stability exercise principles. Current sports medicine reports. 2008, 7(1), 39-44.
- Behm, D. G.; Drinkwater, E. J.; Willardson, J. M.; Cowley, P. M. Canadian Society for Exercise Physiology position stand: The use of instability to train the core in athletic and nonathletic conditioning. Applied Physiology, Nutrition, and Metabolism. 2010, 35(1), 109-112.
- Kline, J. B.; Krauss, J. R.; Maher, S. F.; Qu, X. Core strength training using a combination of home exercises and a dynamic sling system for the management of low back pain in pre-professional ballet dancers: a case series. Journal of dance medicine & science. 2013, 17(1), 24-33.
- Granacher, U.; Gollhofer, A.; Hortobágyi, T.; Kressig, R. W.; Muehlbauer, T. The importance of trunk muscle strength for balance, functional performance, and fall prevention in seniors: a systematic review. Sports medicine. 2013, 43(7), 627-641.
#15. Comments or Suggestions: l. 55-56 "there are many people who do not meet the vaccine investigation for coronavirus" - what is this supposed to mean?
Response: The “Recently, in many countries, there are many people who do not meet the vaccine investigation for coronavirus, and the result of this study can be important at a time when social distancing is essential as the coronavirus is mutating into various types from alpha to delta.” you pointed out was deleted as it was judged not to have much relevance to the purpose of this study.
Methods:
#16. Comments or Suggestions: l. 79-80 explain why these outcomes were selected
Response: Complete blood count (CBC), lymphocytes, and granulocytes selected as outcomes in this study are generally selected variables to observe changes in immune cells.
#17. Comments or Suggestions: l. 114 add manufacturer and data on validity and test-retesr
Response: According your suggestion, we inserted a precise information as follows.….. Smedley dynamometer (TKK-5401, Takei Inc., Tokyo, Japan). Moreover, there are countless studies related to the dynamometer, and the abstract of one research paper is as follows.
[20] Bohannon RW. Muscle strength: clinical and prognostic value of hand-grip dynamometry. Curr Opin Clin Nutr Metab Care. 2015 Sep;18(5):465-70.
Abstract
Purpose of review: Grip strength measured by dynamometry is well established as an indicator of muscle status, particularly among older adults. This review was undertaken to provide a synopsis of recent literature addressing the clinical and prognostic value of hand-grip dynamometry.
Recent findings: Numerous large-scale normative grip strength projects have been published lately. Other recent studies have reinforced the concurrent relationship of grip strength with measures of nutritional status or muscle mass and measures of function and health status. Studies published in the past few years have confirmed the value of grip strength as a predictor of mortality, hospital length of stay, and physical functioning.
Summary: As a whole, the recent literature supports the use of hand-grip dynamometry as a fundamental element of the physical examination of patients, particularly if they are older adults.
#18. Comments or Suggestions: l. 117 sit-up test for 60 sec - add reference and data on validity and test-retest
Response: According to your comment, we suggested the reference as below and attached the reference number in the end of sentence.
[21] Bianco, A.; Jemni, M.; Thomas, E.; Patti, A.; Paoli, A.; Roque, J.R.; Palma, A.; Mammina, C.; Tabacchi, G. A systematic review to determine reliability and usefulness of the field-based test batteries for the assessment of physical fitness in adolescents - The ASSO Project. Int J Occup Med Environ Health. 2015, 28, 445-478.
#19. Comments or Suggestions: l. 145-146 By referring to various literatures - add reference specific to plank exercise
Response: According to your comment, we suggested the reference as below and attached the reference number in the end of sentence.
[24] Park, D.J.; Park, S.Y. Which trunk exercise most effectively activates abdominal muscles? A comparative study of plank and isometric bilateral leg raise exercises. J Back Musculoskelet Rehabil. 2019, 32, 797-802.
[25] Calatayud, J.; Casaña, J.; Martín, F.; Jakobsen, M.D.; Colado, J.C.; Gargallo, P.; Juesas.; Muñoz, V.; Andersen, L.L. Trunk muscle activity during different variations of the supine plank exercise. Musculoskelet Sci Pract. 2017, 28, 54-58.
[26] Choi, J.-H.; Kim, D.-E.; Cynn, H.-S. Comparison of Trunk Muscle Activity Between Traditional Plank Exercise and Plank Exercise With Isometric Contraction of Ankle Muscles in Subjects With Chronic Low Back Pain. J. Strength Cond. Res. 2019. doi: 10.1519/JSC.0000000000003188. Online ahead of print.
#20. Comments or Suggestions: l. 152-154 it is absolutely necessary to provide more details on the progression of the training. Plank exercise is indeed very taxing, and it is of interest how the participant was able to perform at the beginning and how he progressed. Please give details in minutes, not in seconds. It would have been of much interest to have more than two data assessment time points.
Response: According to your comment, we suggested the recorded Tables as below and described more details on the progression of the training on Line 157 to 159.
|
Days |
Sets |
RPE goal |
Week 1 |
Week 2 |
Week 3 |
Week 4 |
||||
|
Time to reach target RPE |
Resting time |
Time to reach target RPE |
Resting time |
Time to reach target RPE |
Resting time |
Time to reach target RPE |
Resting time |
|||
|
Day1 |
1 set |
15 |
67.0 |
60.0 |
62.0 |
60.0 |
71.0 |
60.0 |
75.0 |
60.0 |
|
15 |
65.0 |
60.0 |
65.0 |
60.0 |
70.0 |
60.0 |
74.0 |
60.0 |
||
|
15 |
64.0 |
60.0 |
61.0 |
60.0 |
68.0 |
60.0 |
78.0 |
60.0 |
||
|
15 |
66.0 |
60.0 |
59.0 |
60.0 |
66.0 |
60.0 |
72.0 |
60.0 |
||
|
15 |
59.0 |
60.0 |
59.0 |
60.0 |
62.0 |
60.0 |
66.0 |
60.0 |
||
|
15 |
58.0 |
60.0 |
58.0 |
60.0 |
0.0 |
0.0 |
0.0 |
0.0 |
||
|
2 set |
16 |
84.0 |
60.0 |
82.0 |
60.0 |
92.0 |
60.0 |
96.0 |
60.0 |
|
|
16 |
83.0 |
60.0 |
83.0 |
60.0 |
90.0 |
60.0 |
94.0 |
60.0 |
||
|
16 |
75.0 |
60.0 |
83.0 |
60.0 |
87.0 |
60.0 |
91.0 |
60.0 |
||
|
16 |
74.0 |
60.0 |
76.0 |
60.0 |
85.0 |
60.0 |
88.0 |
60.0 |
||
|
16 |
70.0 |
60.0 |
70.0 |
60.0 |
84.0 |
60.0 |
0.0 |
0.0 |
||
|
3 set |
17 |
105.0 |
60.0 |
99.0 |
60.0 |
116.0 |
60.0 |
121.0 |
60.0 |
|
|
17 |
108.0 |
60.0 |
101.0 |
60.0 |
112.0 |
60.0 |
118.0 |
60.0 |
||
|
17 |
101.0 |
60.0 |
94.0 |
60.0 |
105.0 |
60.0 |
98.0 |
60.0 |
||
|
  |
Total |
  |
1079.0 |
840.0 |
1052.0 |
840.0 |
1108.0 |
780.0 |
1071.0 |
720.0 |
|
Day2 |
1 set |
15 |
69.0 |
60.0 |
67.0 |
60.0 |
74.0 |
60.0 |
78.0 |
60.0 |
|
15 |
67.0 |
60.0 |
65.0 |
60.0 |
73.0 |
60.0 |
76.0 |
60.0 |
||
|
15 |
64.0 |
60.0 |
68.0 |
60.0 |
71.0 |
60.0 |
77.0 |
60.0 |
||
|
15 |
66.0 |
60.0 |
70.0 |
60.0 |
68.0 |
60.0 |
65.0 |
60.0 |
||
|
15 |
60.0 |
60.0 |
72.0 |
60.0 |
66.0 |
60.0 |
68.0 |
60.0 |
||
|
15 |
61.0 |
60.0 |
0.0 |
0.0 |
0.0 |
0.0 |
0.0 |
0.0 |
||
|
2 set |
16 |
75.0 |
60.0 |
77.0 |
60.0 |
74.0 |
60.0 |
77.0 |
60.0 |
|
|
16 |
77.0 |
60.0 |
79.0 |
60.0 |
78.0 |
60.0 |
79.0 |
60.0 |
||
|
16 |
75.0 |
60.0 |
75.0 |
60.0 |
76.0 |
60.0 |
81.0 |
60.0 |
||
|
16 |
74.0 |
60.0 |
79.0 |
60.0 |
74.0 |
60.0 |
79.0 |
60.0 |
||
|
16 |
68.0 |
60.0 |
72.0 |
60.0 |
65.0 |
60.0 |
73.0 |
60.0 |
||
|
3 set |
17 |
101.0 |
60.0 |
102.0 |
60.0 |
121.0 |
60.0 |
118.0 |
60.0 |
|
|
17 |
95.0 |
60.0 |
110.0 |
60.0 |
115.0 |
60.0 |
115.0 |
60.0 |
||
|
17 |
97.0 |
60.0 |
98.0 |
60.0 |
96.0 |
60.0 |
99.0 |
60.0 |
||
|
  |
Total |
  |
1049.0 |
840.0 |
1034.0 |
780.0 |
1051.0 |
780.0 |
1085.0 |
780.0 |
|
Day3 |
1 set |
15 |
71.0 |
60.0 |
71.0 |
60.0 |
71.0 |
60.0 |
76.0 |
60.0 |
|
15 |
67.0 |
60.0 |
73.0 |
60.0 |
73.0 |
60.0 |
75.0 |
60.0 |
||
|
15 |
64.0 |
60.0 |
67.0 |
60.0 |
75.0 |
60.0 |
67.0 |
60.0 |
||
|
15 |
68.0 |
60.0 |
68.0 |
60.0 |
72.0 |
60.0 |
66.0 |
60.0 |
||
|
15 |
60.0 |
60.0 |
64.0 |
60.0 |
70.0 |
60.0 |
67.0 |
60.0 |
||
|
15 |
57.0 |
60.0 |
0.0 |
0.0 |
0.0 |
0.0 |
0.0 |
0.0 |
||
|
2 set |
16 |
73.0 |
60.0 |
73.0 |
60.0 |
88.0 |
60.0 |
91.0 |
60.0 |
|
|
16 |
77.0 |
60.0 |
75.0 |
60.0 |
85.0 |
60.0 |
89.0 |
60.0 |
||
|
16 |
70.0 |
60.0 |
74.0 |
60.0 |
84.0 |
60.0 |
88.0 |
60.0 |
||
|
16 |
71.0 |
60.0 |
71.0 |
60.0 |
78.0 |
60.0 |
82.0 |
60.0 |
||
|
16 |
65.0 |
60.0 |
69.0 |
60.0 |
69.0 |
60.0 |
0.0 |
0.0 |
||
|
3 set |
17 |
99.0 |
60.0 |
112.0 |
60.0 |
115.0 |
60.0 |
119.0 |
60.0 |
|
|
17 |
95.0 |
60.0 |
108.0 |
60.0 |
99.0 |
60.0 |
109.0 |
60.0 |
||
|
17 |
89.0 |
60.0 |
98.0 |
60.0 |
87.0 |
60.0 |
100.0 |
60.0 |
||
|
  |
Total |
  |
1026.0 |
840.0 |
1023.0 |
780.0 |
1066.0 |
780.0 |
1029.0 |
720.0 |
|
Day4 |
1 set |
15 |
69.0 |
60.0 |
69.0 |
60.0 |
69.0 |
60.0 |
75.0 |
60.0 |
|
15 |
65.0 |
60.0 |
69.0 |
60.0 |
77.0 |
60.0 |
78.0 |
60.0 |
||
|
15 |
64.0 |
60.0 |
71.0 |
60.0 |
76.0 |
60.0 |
78.0 |
60.0 |
||
|
15 |
68.0 |
60.0 |
68.0 |
60.0 |
68.0 |
60.0 |
72.0 |
60.0 |
||
|
15 |
66.0 |
60.0 |
67.0 |
60.0 |
76.0 |
60.0 |
74.0 |
60.0 |
||
|
15 |
61.0 |
60.0 |
0.0 |
0.0 |
0.0 |
0.0 |
0.0 |
0.0 |
||
|
2 set |
16 |
74.0 |
60.0 |
71.0 |
60.0 |
87.0 |
60.0 |
89.0 |
60.0 |
|
|
16 |
76.0 |
60.0 |
72.0 |
60.0 |
84.0 |
60.0 |
83.0 |
60.0 |
||
|
16 |
71.0 |
60.0 |
74.0 |
60.0 |
82.0 |
60.0 |
80.0 |
60.0 |
||
|
16 |
71.0 |
60.0 |
68.0 |
60.0 |
75.0 |
60.0 |
78.0 |
60.0 |
||
|
16 |
68.0 |
60.0 |
66.0 |
60.0 |
65.0 |
60.0 |
77.0 |
60.0 |
||
|
3 set |
17 |
98.0 |
60.0 |
106.0 |
60.0 |
110.0 |
60.0 |
116.0 |
60.0 |
|
|
17 |
92.0 |
60.0 |
94.0 |
60.0 |
107.0 |
60.0 |
111.0 |
60.0 |
||
|
17 |
85.0 |
60.0 |
88.0 |
60.0 |
103.0 |
60.0 |
95.0 |
60.0 |
||
|
  |
Total |
  |
1028.0 |
840.0 |
983.0 |
780.0 |
1079.0 |
780.0 |
1106.0 |
780.0 |
|
Day5 |
1 set |
15 |
65.0 |
60.0 |
68.0 |
60.0 |
72.0 |
60.0 |
78.0 |
60.0 |
|
15 |
62.0 |
60.0 |
63.0 |
60.0 |
71.0 |
60.0 |
77.0 |
60.0 |
||
|
15 |
64.0 |
60.0 |
65.0 |
60.0 |
68.0 |
60.0 |
74.0 |
60.0 |
||
|
15 |
62.0 |
60.0 |
62.0 |
60.0 |
66.0 |
60.0 |
70.0 |
60.0 |
||
|
15 |
60.0 |
60.0 |
56.0 |
60.0 |
63.0 |
60.0 |
68.0 |
60.0 |
||
|
15 |
61.0 |
60.0 |
51.0 |
60.0 |
0.0 |
0.0 |
0.0 |
0.0 |
||
|
2 set |
16 |
72.0 |
60.0 |
66.0 |
60.0 |
85.0 |
60.0 |
90.0 |
60.0 |
|
|
16 |
71.0 |
60.0 |
67.0 |
60.0 |
82.0 |
60.0 |
94.0 |
60.0 |
||
|
16 |
71.0 |
60.0 |
65.0 |
60.0 |
72.0 |
60.0 |
91.0 |
60.0 |
||
|
16 |
65.0 |
60.0 |
65.0 |
60.0 |
68.0 |
60.0 |
86.0 |
60.0 |
||
|
16 |
66.0 |
60.0 |
66.0 |
60.0 |
66.0 |
60.0 |
84.0 |
60.0 |
||
|
3 set |
17 |
89.0 |
60.0 |
92.0 |
60.0 |
104.0 |
60.0 |
112.0 |
60.0 |
|
|
17 |
90.0 |
60.0 |
87.0 |
60.0 |
100.0 |
60.0 |
105.0 |
60.0 |
||
|
17 |
87.0 |
60.0 |
78.0 |
60.0 |
82.0 |
60.0 |
104.0 |
60.0 |
||
|
  |
Total |
  |
985.0 |
840.0 |
951.0 |
840.0 |
999.0 |
780.0 |
1133.0 |
780.0 |
The above table recorded for one month is summarized and shown in the table below.
|
Weeks |
Days |
Plank exercise |
Resting time |
Total (min) |
||
|
Seconds |
Minute |
Seconds |
Minute |
|||
|
Week 1 |
Day 1 |
1079.00 |
17.98 |
840.00 |
14 |
31.98 |
|
Day 2 |
1049.00 |
17.48 |
840.00 |
14 |
31.48 |
|
|
Day 3 |
1026.00 |
17.10 |
840.00 |
14 |
31.10 |
|
|
Day 4 |
1028.00 |
17.13 |
840.00 |
14 |
31.13 |
|
|
Day 5 |
985.00 |
16.42 |
840.00 |
14 |
30.42 |
|
|
Mean |
  |
17.22 |
  |
14.00 |
31.22 |
|
|
Week 2 |
Day 1 |
1052.00 |
17.53 |
840.00 |
14 |
31.53 |
|
Day 2 |
1034.00 |
17.23 |
780.00 |
13 |
30.23 |
|
|
Day 3 |
1023.00 |
17.05 |
780.00 |
13 |
30.05 |
|
|
Day 4 |
983.00 |
16.38 |
780.00 |
13 |
29.38 |
|
|
Day 5 |
951.00 |
15.85 |
840.00 |
14 |
29.85 |
|
|
Mean |
  |
16.81 |
  |
13.40 |
30.21 |
|
|
Week 3 |
Day 1 |
1108.00 |
18.47 |
780.00 |
13 |
31.47 |
|
Day 2 |
1051.00 |
17.52 |
780.00 |
13 |
30.52 |
|
|
Day 3 |
1066.00 |
17.77 |
780.00 |
13 |
30.77 |
|
|
Day 4 |
1079.00 |
17.98 |
780.00 |
13 |
30.98 |
|
|
Day 5 |
999.00 |
16.65 |
780.00 |
13 |
29.65 |
|
|
Mean |
  |
17.68 |
  |
13.00 |
30.68 |
|
|
Week 4 |
Day 1 |
1071.00 |
17.85 |
720.00 |
12 |
29.85 |
|
Day 2 |
1085.00 |
18.08 |
780.00 |
13 |
31.08 |
|
|
Day 3 |
1029.00 |
17.15 |
720.00 |
12 |
29.15 |
|
|
Day 4 |
1106.00 |
18.43 |
780.00 |
13 |
31.43 |
|
|
Day 5 |
1130.00 |
18.83 |
780.00 |
13 |
31.83 |
|
|
Mean |
  |
18.07 |
  |
12.6 |
30.67 |
|
In the table above, in consideration of the amount of text, only the description was put in the text without inserting it. The contents inserted into the text are as follows.
“The average plank exercise time performed by the subjects in the first week was 17.22 min, and the rest time was 14 min, making the program a total of 31.22 min. On the other hand, the average plank exercise time of week 2 was 16.81 min, and the rest time was 13.4 min, taking a total of 30.21 min. The average plank exercise time of the 3rd week was 17.68 min, and the rest time was 13 min, a total of 30.68 min. Finally, the average plank exercise time of week 4 was 18.07 min, and the rest time was 12.6 min, which took a total of 30.67 min. That is, in this study, the set time at RPE 15 and RPE 16 was adjusted to match the total program time to around 30 min, and the time at RPE 17 was not adjusted for high-intensity plank exercise.”
Descriptions on Line 157 to 159 from original manuscript are as follows.
Elbow plank exercise was performed for the study. Lying stretches were performed for 5 min before and after the plank exercise. In the work-out phase, the participant maintained a straight, strong line from head to toes with no lowering of the hips with the shoulders and elbows flexed at 90° as shown in Figure. 1.
In this study, Borg's 20-scale RPE chart for deciding his plank intensity was copied to the size of the palm of the hand and provided to the participant. By referring to various literatures [ ], the plank exercise was conducted with the goal of reaching extremely hard intensity. The exercise intensities for elbow plank were ranged from RPE 15 (feeling hard) and RPE 17 (feeling extremely hard). Each 30-min session was divided into three 10-min stages, each with increasing intensity. In other words, elbow plank exercises were performed until reaching RPE 15 (hard) in the first 10 min, RPE 16 (hard to very hard) in the second 10 min, and RPE 17 (very hard) in the third 10 min. The same posture was maintained until reaching the target RPE for each stage. When the target RPE was reached, the subject took rested for 1 min and then restarted throughout the approximate 30-min duration. In other words, if each intensity was exceeded and the correct plank posture was not achieved, 1 min was allowed to rest then to re-execute.
The average plank exercise time performed by the subjects in the first week was 17.22 min, and the rest time was 14 min, making the program a total of 31.22 min. On the other hand, the average plank exercise time of week 2 was 16.81 min, and the rest time was 13.4 min, taking a total of 30.21 min. The average plank exercise time of the 3rd week was 17.68 min, and the rest time was 13 min, a total of 30.68 min. Finally, the average plank exercise time of week 4 was 18.07 min, and the rest time was 12.6 min, which took a total of 30.67 min. That is, in this study, the set time at RPE 15 and RPE 16 was adjusted to match the total program time to around 30 min, and the time at RPE 17 was not adjusted for high-intensity plank exercise.
Results:
#21. Comments or Suggestions: Table 4. the change in flexibility does not seem realistic.
Response: The 'flexibility' presented in Table 4 was previously increased from -6 cm to 3 cm, so the expression 'improved' was included in the text.
Discussion:
#22. Comments or Suggestions: l. 202 quality of life is not the subject matter of this report, and the statement as such is not true. Please delete or rewrite in a manner that is it true, provide reference.
Response: We decide to delete that sentence based on your comments. For example: Physical exercise has a wide range of benefits during and after intervention that in turn positively influence the quality of life in everyone.
#23. Comments or Suggestions: l. 220 "Previous studies have found both positive and negative effects of regular exercise in patients with various types of diseases". - this sentence is empty, and unscientific. Please be precise and use appropriate references
Response: We decide to delete that sentence based on your comments. For example: Previous studies have found both positive and negative effects of regular exercise in patients with various types of diseases.
#24. Comments or Suggestions: l 223 add reference
Response: According to your comment, we suggested the reference as below and attached the reference number in the end of sentence.
…Recently, the coronavirus has disrupted the immunocytes of many people as a result from mid to long-term isolation [28].
[28] Nyenhuis, S.M.; Greiwe, J.; Zeiger, J.S.; Nanda, A.; Cooke, A. Exercise and Fitness in the age of social distancing during the COVID-19 Pandemic. J Allergy Clin Immunol Pract, 2020, 8, 2152-2155.
#25. Comments or Suggestions: l. 289 add reference
Response: According to your comment, we suggested the reference as below and attached the reference number in the end of sentence.
…when the number of NK cells increases, cytotoxicity also increases, but cytotoxicity decreases when there is a specific cause, such as having cancer cells, when exposed to excessive stress, or when exercising excessively [42].
[42] Shephard, R.J.; Shek, P.N.; DiNubile, N.A. Exercise, immunity, and susceptibility to infection: a j-shaped relationship? Phys Sportsmed. 1999, 27, 47-71.
#26. Comments or Suggestions: Whole of the discussion: please make sure to differentiate between endurance vs. strength training in terms of immune response.
Response: According to your comment, we made sure to differentiate between endurance vs. strength training in terms of immune response. For example, … In general, it is known that the functions of B cells are suppressed after intense, long-duration exercise, while lymphocyte concentrations have been shown to increase during acute exercise and fall below pre-workout values after long-term endurance exercise [27,38]. ……. Strenuous exercise, but not moderate exercise, is followed by a decreased concentration of lymphocytes in the bloodstream, which results in low lymphocyte levels in tissues [41].
Limitations:
#27. Comments or Suggestions: A limitation of the study is the questionable generalisability of this protocol. Plank exercise is indeed very challenging, and if performed incorrectly can induce low back pain or shoulder pain. This should be also discussed.
Response: According to your comments, we inserted below sentence to the limitation as follows.
…. Third, a careful approach is required because plank exercises performed at less than moderate intensity or high intensity can cause shoulder joint or back pain.
We sincerely appreciate your comments and suggestion.
Best regards,

Reviewer 2 Report
It is an interesting article concerning a practical solution of an important problem concerning a large amount of peoples.
It would be convenient a more detailed description of the subject you studied: physical activity and sport habits, detailed metabolic evaluation, previous diseases and risk profile.
In the tables you better indicate the significant level, after description of statystical analysis in the methods.
Author Response
Answers to 2nd reviewer’s comments
Thank you for your kind advice and comments for publication in Medicina. We revised our manuscript as per your comments. We represented the specific modifications in response to the comments by blue-letters in our manuscript. We sincerely appreciate your comments because your comments make our manuscript better. Details of responses about reviewer’s comments are as follows:
#1. Comments or Suggestions: It is an interesting article concerning a practical solution of an important problem concerning a large amount of peoples.
Response: We are very grateful for your positive evaluation of our thesis.
#2. Comments or Suggestions: It would be convenient a more detailed description of the subject you studied: physical activity and sport habits, detailed metabolic evaluation, previous diseases and risk profile.
Response: According to your suggestion, we added to more specific information about the participant on Line 63 as follows: He played basketball on a regular basis 10 years ago, but now his main physical activity is only walking for 30 min twice a day. He had no cardiac or metabolic disease, but his father had hypertension. He has been smoking a pack of cigarettes a day and drank moderately about three days a week. He had not been taking any medication. Of course, we recommended abstaining from exercise, smoking and alcohol consumption for one month during the experiment.
#3. Comments or Suggestions: In the tables you better indicate the significant level, after description of statystical analysis in the methods.
Response: You are right, but our thesis submitted is a 'Case report' for one subject. Therefore, data processing by statistical analysis is not possible, so please understand it in a broad sense.
We sincerely appreciate your comments and suggestion.
Best regards,

Round 2
Reviewer 1 Report
Thank you for the improvements. Two minor questions:
How did you make sure the participant didn´t change his physical activity during the study? (i.e. that he didn´t perform other activities than the plank exercise?)
Please make sure to include the table with the detailed exercise data (Exercise and rest times during the 4 weeks) into the manuscript.
Author Response
Answers to 1st reviewer’s comments (2nd Revision)
Thank you for your kind advice and comments for publication in Medicina. We re-revised our manuscript as per your comments. We represented the specific modifications in response to the comments by blue-letters in our manuscript. We sincerely appreciate your comments because your comments make our manuscript better. Details of responses about reviewer’s comments are as follows:
#1. Comments or Suggestions: How did you make sure the participant didn´t change his physical activity during the study? (i.e. that he didn´t perform other activities than the plank exercise?)
Response: One participant in this study was a person doing similar work in the same school laboratory. So, we did the plank in the same pattern for a month, but controlled not to do any other activity. Thanks for the questions and comments.
#2. Comments or Suggestions: Please make sure to include the table with the detailed exercise data (Exercise and rest times during the 4 weeks) into the manuscript.
Response: Following the reviewer's recommendation, we provide detailed exercise data (Exercise and rest times during the 4 weeks) into the manuscript. The structure of the inserted table is as follows.
Table 1. Plank exercise time and resting time for the participant
|
Days |
Plank exercise |
Resting time |
Total (minutes) |
|||
|
seconds |
minutes |
seconds |
minutes |
|||
|
Week 1 |
Day 1 |
1079.00 |
17.98 |
840.00 |
14 |
31.98 |
|
Day 2 |
1049.00 |
17.48 |
840.00 |
14 |
31.48 |
|
|
Day 3 |
1026.00 |
17.10 |
840.00 |
14 |
31.10 |
|
|
Day 4 |
1028.00 |
17.13 |
840.00 |
14 |
31.13 |
|
|
Day 5 |
985.00 |
16.42 |
840.00 |
14 |
30.42 |
|
|
Mean |
  |
17.22 |
  |
14.00 |
31.22 |
|
|
Week 2 |
Day 1 |
1052.00 |
17.53 |
840.00 |
14 |
31.53 |
|
Day 2 |
1034.00 |
17.23 |
780.00 |
13 |
30.23 |
|
|
Day 3 |
1023.00 |
17.05 |
780.00 |
13 |
30.05 |
|
|
Day 4 |
983.00 |
16.38 |
780.00 |
13 |
29.38 |
|
|
Day 5 |
951.00 |
15.85 |
840.00 |
14 |
29.85 |
|
|
Mean |
  |
16.81 |
  |
13.40 |
30.21 |
|
|
Week 3 |
Day 1 |
1108.00 |
18.47 |
780.00 |
13 |
31.47 |
|
Day 2 |
1051.00 |
17.52 |
780.00 |
13 |
30.52 |
|
|
Day 3 |
1066.00 |
17.77 |
780.00 |
13 |
30.77 |
|
|
Day 4 |
1079.00 |
17.98 |
780.00 |
13 |
30.98 |
|
|
Day 5 |
999.00 |
16.65 |
780.00 |
13 |
29.65 |
|
|
Mean |
  |
17.68 |
  |
13.00 |
30.68 |
|
|
Week 4 |
Day 1 |
1071.00 |
17.85 |
720.00 |
12 |
29.85 |
|
Day 2 |
1085.00 |
18.08 |
780.00 |
13 |
31.08 |
|
|
Day 3 |
1029.00 |
17.15 |
720.00 |
12 |
29.15 |
|
|
Day 4 |
1106.00 |
18.43 |
780.00 |
13 |
31.43 |
|
|
Day 5 |
1130.00 |
18.83 |
780.00 |
13 |
31.83 |
|
|
Mean |
  |
18.07 |
  |
12.6 |
30.67 |
|
We sincerely appreciate your comments and suggestion.
Best regards,
